# Glucose Fluctuation and Severe Internal Carotid Artery Siphon Stenosis in Type 2 Diabetes Patients

**DOI:** 10.3390/nu13072379

**Published:** 2021-07-12

**Authors:** Futoshi Eto, Kazuo Washida, Masaki Matsubara, Hisashi Makino, Akio Takahashi, Kotaro Noda, Yorito Hattori, Yuriko Nakaoku, Kunihiro Nishimura, Kiminori Hosoda, Masafumi Ihara

**Affiliations:** 1Department of Neurology, National Cerebral and Cardiovascular Center, Suita 564-8565, Japan; eto.futoshi@gmail.com (F.E.); oremonox@gmail.com (A.T.); noda.kotaro@ncvc.go.jp (K.N.); yoh2019@ncvc.go.jp (Y.H.); ihara@ncvc.go.jp (M.I.); 2Division of Diabetes and Lipid Metabolism, National Cerebral and Cardiovascular Center, Suita 564-8565, Japan; matsubara.m@ncvc.go.jp (M.M.); makinoh@ncvc.go.jp (H.M.); kiminorihosoda@ncvc.go.jp (K.H.); 3Department of Preventive Medicine and Epidemiology, National Cerebral and Cardiovascular Center, Suita 564-8565, Japan; yurikon@ncvc.go.jp (Y.N.); knishimu@ncvc.go.jp (K.N.)

**Keywords:** continuous glucose monitoring, glucose fluctuation, intracranial artery stenosis, mean amplitude of glycemic excursions, standard deviation

## Abstract

The impact of glucose fluctuation on intracranial artery stenosis remains to be elucidated. This study aimed to investigate the association between glucose fluctuation and intracranial artery stenosis. This was a cross-sectional study of type 2 diabetes mellitus (T2DM) patients equipped with the FreeStyle Libre Pro continuous glucose monitoring system (Abbott Laboratories) between February 2019 and June 2020. Glucose fluctuation was evaluated according to the standard deviation (SD) of blood glucose, coefficient of variation (%CV), and mean amplitude of glycemic excursions (MAGE). Magnetic resonance angiography was used to evaluate the degree of intracranial artery stenosis. Of the 103 patients, 8 patients developed severe internal carotid artery (ICA) siphon stenosis (≥70%). SD, %CV, and MAGE were significantly higher in the severe stenosis group than in the non-severe stenosis group (<70%), whereas there was no significant intergroup difference in the mean blood glucose and HbA1c. Multivariable logistic regression analysis adjusted for sex showed that SD, %CV, and MAGE were independent factors associated with severe ICA siphon stenosis. In conclusion, glucose fluctuation is significantly associated with severe ICA siphon stenosis in T2DM patients. Thus, glucose fluctuation can be a target of preventive therapies for intracranial artery stenosis and ischemic stroke.

## 1. Introduction

It was estimated that 451 million individuals globally have diabetes mellitus (DM) in 2017 [1]. DM is a major cause of blindness, kidney failure, heart attacks, stroke, and lower limb amputation. Long-term management of DM can prevent atherosclerotic cardiovascular disease (ASCVD) such as ischemic stroke or acute coronary syndrome. Patients with severe intracranial artery stenosis have the highest rate of recurrent stroke [2,3]. There have been various reports on the relationship between DM and intracranial artery stenosis [4,5], but the findings have been conflicting. Although some studies reported that elevated hemoglobin A1c (HbA1c) and fasting blood glucose levels are associated with intracranial artery stenosis [6], others showed no correlations [7]. Thus, the usefulness of HbA1c and fasting blood glucose levels as predictors of intracranial artery stenosis remains unclear.

There are various indicators for blood glucose control in patients with DM; these include hemoglobin A1c (HbA1c) and glycoalbumin. However, these indicators only reflect the mean blood glucose level for a certain period and cannot reflect glucose fluctuation. Atherosclerotic stenosis such as intracranial artery stenosis and coronary artery stenosis are major complications of DM. Glucose fluctuation can cause atherosclerosis because it induces chronic inflammation and oxidative stress in the vasculature [8]. Thus, prevention of atherosclerosis in patients with DM requires targeting glucose fluctuation. Continuous glucose monitoring (CGM) systems, such as the FreeStyle Libre Pro, have been recently approved for use in clinical practice. In contrast to self-monitoring of blood glucose (SMBG) where up to 80% of hypoglycemia and hyperglycemia can be missed [9], CGM enables a continuous monitoring of blood glucose levels and fluctuations.

Recent clinical studies have shown that blood glucose fluctuation is related to ASCVD [7,10,11,12]. Furthermore, glucose fluctuation could predict prognosis after acute coronary syndrome [13]. However, although blood glucose fluctuation is associated with the risk of many cardiovascular diseases, the relationship between blood glucose fluctuation and intracranial artery stenosis remains unclear. Therefore, this study aimed to investigate the relationship between glucose fluctuation and intracranial artery stenosis in type 2 DM (T2DM) patients who are using the FreeStyle Libre Pro continuous glucose monitoring system.

## 2. Materials and Methods

### 2.1. Study Design and Patients

This retrospective, observational, cross-sectional study was performed at the National Cerebral and Cardiovascular Center (NCVC), Suita, Osaka, Japan. This study is part of an ongoing prospective longitudinal study on the relationship between glucose fluctuation and cognitive function in T2DM (PROPOSAL Study: Trial Registration, University Hospital Medical Information Network Clinical Trial Registry (UMIN000038546)) [14].

T2DM patients with mild cognitive impairment (MCI) were enrolled in the registry between February 2019 and June 2020. The PROPOSAL Study is aimed at evaluating the relationships between glucose fluctuation indices assessed by CGM and cognitive function among elderly patients with T2DM. Therefore, patients are limited to those aged 65–85 years. T2DM was diagnosed according to the Japan Diabetes Society criteria. MCI was diagnosed based on the clinical course and a score of 17–25 on the Japanese version of Montreal Cognitive Assessment scale [14,15,16]. Carotid artery stenosis was evaluated according to the North American Symptomatic Carotid Endarterectomy Trial method [17]. Patients with ≥80% carotid artery stenosis [18] or those undergoing renal replacement therapy [19] were excluded because these conditions could affect cognitive function. Additionally, those taking antidementia drugs or having underlying comorbidities affecting cognitive function (depression, thyroid dysfunction, and vitamin B1, vitamin B12, and folate deficiency) were excluded. Sex, age, baseline patient characteristics including current smoking status, medical history such as hypertension or active use of antihypertensive medications, dyslipidemia or active use of lipid-lowering agents, T2DM or antidiabetic treatment, atrial fibrillation or antidiabetic treatment, medical history of percutaneous coronary intervention or coronary artery bypass grafting (PCI/CABG), and former ischemic stroke episode, were collected from the registry.

All subjects gave their informed consent for inclusion before they participated in the study. The study was conducted in accordance with the Declaration of Helsinki, and the protocol was approved by the Ethics Committee of NCVC (Project identification code M30-110-3).

### 2.2. Imaging Protocol

Magnetic resonance imaging (MRI) was performed with a 3-Tesla system. The vessels constituting the intracranial artery were defined as shown in Figure 1: (i) the A1 or A2 segment of the anterior cerebral arteries (ACA), (ii) the C1 to C5 segment of the intracranial internal carotid arteries (ICA) categorized according to Fischer’s classification [20], (iii) the P1 or P2 segment of the posterior cerebral arteries (PCA), and (iv) the M1 or M2 segment of the middle cerebral arteries (MCA).

Magnetic resonance angiography (MRA) findings of the vessels constituting the intracranial artery were independently read by two stroke neurologists (F.E. and A.T.) blinded to the clinical information, to determine the anatomical variations. Disagreements were resolved through a joint assessment until consensus was reached. The percentage of stenosis for each vessel was listed in 5% increments.

Percent stenosis was measured using the Warfarin-Aspirin Symptomatic Intracranial Disease (WASID) method [21]. The percentage was calculated by MRA using a previous method as follows: (1) the most severe stenosis spot on the maximum-intensity projection or axial source images was measured using the time-of-flight method; then, (2) we measured at the widest, non-tortuous, normal portion of the petrous ICA parallel to the site of stenosis [22]. Intracranial artery stenosis was evaluated on the side with the stronger stenosis. The degree of stenosis was categorized into two categories as severe stenosis (i.e., ≥70% stenosis [3] at specific segments of the intracranial artery: A1 or A2 segment of the ACA, C1-C5 segment of the ICA, P1 or P2 segment of the PCA, and M1 or M2 segment of the MCA) and non-severe stenosis (i.e., <70% stenosis), based on a previous report [23].

### 2.3. Continuous Glucose Monitoring

The FreeStyle Libre Pro continuous glucose monitoring (FLP-CGM) system (Abbott Laboratories, Chicago, IL, USA) is an interstitial glucose monitoring device with an established accuracy [24]. The FLP sensor is disposable and inserted on the back of an upper extremity for up to 14 days. A unique feature of the sensor is that calibration is not required using SMBG, and after it is removed, data can be downloaded, and glucose profiles evaluated. In this study, the mean glucose, standard deviation (SD), percent coefficient of variation (%CV) [25], and the mean amplitude of glycemic excursions (MAGE) [26] were calculated to evaluate glucose fluctuation. Considering the concerns about the lack of the accuracy of the date at day 1 [24], we used the data from day 2 to the end of recording (maximally, day 14). Additionally, patients in whom blood glucose fluctuations were not measured by CGM within 7 days were excluded according to former protocols [27].

### 2.4. Statistical Analyses

Continuous variables are shown as the mean ± standard deviation and compared using a *t*-test if data were normally distributed. Meanwhile, categorical variables are shown as frequencies and percentages and compared using Fisher’s exact test. Agreement in stenosis assessments between the two physicians was assessed using weighted kappa statistics. These statistics are appropriate when there are more than two ordered categories and adjust for chance agreement and degree of disagreement between raters. Logistic regression models were used to evaluate the associations of each glucose fluctuation factor with severe and non-severe intracranial stenosis. Univariable logistic regression models were used to calculate odds ratios (ORs) and 95% confidence intervals (CIs). Sex, age, current smoking, duration years of T2DM, medical history of hypertension, dyslipidemia, atrial fibrillation, former ischemic stroke episode, and PCI/CABG were entered in the univariable models. Multivariable logistic regression analyses were performed using covariates significantly associated with intracranial stenosis in univariable models.

All statistical analyses were conducted by two physicians (F.E. and Y.N.) using JMP 14.0.0 statistical software (SAS Institute Inc., Cary, NC, USA) and Stata 15.1 software (StataCorp, College Station, TX, USA). A *p* value of <0.05 was considered statistically significant.

## 3. Results

### 3.1. Baseline Patient Characteristics

The patient inclusion flow chart is shown in Figure 2. Of the 109 T2DM patients enrolled in the registry, 103 patients with a mean age of 76 ± 5 years (females, 30%) were included in the current analysis. Six patients were excluded due to missing baseline data (*n* = 4), non-availability for MRI (*n* = 1), and evaluation with 1.5 Tesla system for MRI (*n* = 1). CGM data of all 103 patients were obtained.

The patient’s baseline characteristics stratified according to the WASID method are shown in Table 1.

### 3.2. Head Magnetic Resonance Angiography Findings

Of the 103 patients examined, 8 patients presented with severe (≥70%) ICA siphon stenosis (severe stenosis group: 8%), while 95 patients presented non-severe (<70%) ICA siphon stenosis (non-severe stenosis group: 92%). Among these 95 patients, 48 and 47 patients had moderate (50–70%) and mild (0–50%) ICA siphon stenosis, respectively. A representative case of severe stenosis of the left ICA siphon on MRA is shown in Figure 3.

Except ICA siphon, severe stenoses (≥70%) were not observed in any other intracranial arteries. In addition, moderate stenoses (50–70%) were only observed at the M1 portion of the MCA in 3 of the 103 patients. Regarding the consistency of intracranial artery stenosis evaluation, the inter-rater agreement of the quadratic weighted kappa statistic for the evaluation of the vessels constituting the intracranial artery was 0.952, indicating high consistency.

### 3.3. Association between Glucose Fluctuation and Intracranial Artery Stenosis

Compared with the non-severe stenosis group (*n* = 95), the severe stenosis group (*n* = 8) showed significantly higher variability in the three indices of glucose fluctuation: SD (53 mg/dL vs. 39 mg/dL, *p <* 0.01), %CV (36 vs. 29, *p <* 0.01), and MAGE (114 mg/dL vs. 90 mg/dL, *p <* 0.01) (Table 1). Meanwhile, other vascular risk factors, such as smoking, hypertension, dyslipidemia, mean blood glucose, HbA1c, and duration years of T2DM, were not significantly different between the two groups (Table 1). Scatter plots showing the relationships between glucose fluctuation and ICA siphon stenosis are shown in Figure 4.

In univariable analysis, the severe stenosis group showed significantly higher SD (OR, 3.60; 95% CI, 1.60–8.08; *p <* 0.01), %CV (OR, 7.85; 95% CI, 1.90–32.5; *p <* 0.01), and MAGE (OR, 1.56; 95% CI, 1.11–2.20; *p =* 0.01). Multivariable logistic regression analysis showed that these factors remained significantly associated with severe ICA siphon stenosis after adjustment for sex (SD: OR, 3.00; 95% CI, 1.32–6.84; *p <* 0.01; %CV: OR, 5.55; 95% CI, 1.23–25.2; *p =* 0.03; and MAGE: OR, 1.52; 95% CI, 1.06–2.19; *p =* 0.02) (Table 2).

As for the cases with moderate (50–70%) and mild (0–50%) ICA siphon stenosis, there was no significant intergroup difference in all variables including SD, %CV, and MAGE (Appendix A). There was also no significant intergroup difference in all variables including SD, %CV, and MAGE for the cases with moderate (50–70%) and mild (0–50%) MCA M1 stenosis (Appendix A).

## 4. Discussion

The relationship between blood glucose fluctuation and intracranial artery stenosis in T2DM patients remains unclear. In this study, patients with severe ICA siphon stenosis had higher blood glucose fluctuations as assessed with SD, %CV, and MAGE. Meanwhile, there were no significant differences for other vascular risk factors, such as hypertension, dyslipidemia, mean blood glucose levels, HbA1c, and duration in years of T2DM. To our best knowledge, this is the first study to reveal the association between intracranial artery stenosis and glucose fluctuation.

There are several possible mechanisms by which blood glucose fluctuation causes the atherosclerotic stenosis of the major intracranial arteries. Atherosclerosis is a complex multifactorial disease and often causes diabetic macrovascular complications. Glucose fluctuation plays a key role in the development of atherosclerosis. One of the most common causative factors for atherosclerosis by glucose fluctuation is an increase in oxidative stress due to a rapid blood glucose change that causes vascular endothelial damage. Compared with chronic sustained hyperglycemia, glucose fluctuations induce a more specific effect on oxidative stress [8]. Severe blood glucose fluctuation is known to lower the number of vascular endothelial progenitor cells [28]. Blood glucose fluctuation is also correlated with carotid intima media thickness (IMT) [29], which is an indicator of subclinical atherosclerosis. Coronary plaque has been reported to be correlated with glucose fluctuation [30].

Additionally, glucose fluctuation usually includes hyperglycemia and hypoglycemia, and these are also associated with the presence and severity of cardiovascular disease in DM patients [31]. Hyperglycemia increases the advanced glycation endproducts (AGEs), and the binding of the AGEs to the receptor of AGEs induces oxidative stress and inflammation via the NF-kappa B pathway, leading to atherosclerosis [32]. Furthermore, a meta-analysis by Liang et al. showed that minimizing glucose fluctuation improved insulin resistance and carotid IMT thickness, thus lowering the risk of cardiovascular disease [33]. Reductions in glucose fluctuation by DPP-IV inhibitors can prevent atherosclerosis progression in T2DM patients by lowering inflammation and oxidative stress [34].

However, there is still limited evidence on the association between glucose fluctuations and cerebrovascular lesions. Glucose is the primary energy source for the brain, and severe glucose fluctuations have been associated with numerous types of central nervous system damage [35]. Several studies demonstrated that oxidative stress and inflammation due to blood glucose fluctuations impair the blood–brain barrier [36] or induce hypercoagulability and suppression of the fibrinolytic system [37]. Blood glucose fluctuations also worsen the progression of cerebral white matter lesions [38] and the prognosis of cerebral infarction [39,40]. Increasing evidence shows that glucose fluctuation significantly increases oxidative stress, leading to neuroinflammation and cognitive dysfunction [35]. However, the detailed mechanisms by which glucose fluctuation causes cerebrovascular lesions remain to be elucidated, highlighting the need for further studies.

With respect to the site of the intracranial artery stenosis, our study showed that intracranial artery stenosis particularly occurred at the siphon portion of the ICA. Few studies have examined the sites of intracranial artery stenosis. Atherosclerotic stenosis often occurs at sites with complex hemodynamics, such as arteries with high curvature or bifurcations. A study on fluid dynamics in morphology identified three preferred sites of stenoses along the carotid siphon with low and highly oscillatory wall shear stress [41]. Another review showed that low and oscillatory shear stress is closely associated with atherogenesis [42]. In our study, stenosis was also found along the carotid siphon area. As for the degree of the intracranial artery stenosis, Mo et al. [7] assessed the relationship between glucose fluctuation and degree of intracranial artery stenosis and found no significant relationship. However, stenosis was defined as more than 50% thickening of the arterial wall, and this could have affected the finding. A previous analysis of the predictors of ischemic stroke in symptomatic intracranial arterial stenosis showed that patients with ≥70% intracranial stenosis have a ≥2 times higher risk of stroke than patients with <70% stenosis [3]. In this context, we compared patients with severe intracranial stenosis (≥70% stenosis) and non-severe intracranial stenosis (<70%), and found a significant difference in glucose fluctuation between them.

In this study, the proportion of female patients was higher in the severe ICA siphon stenosis group than in the non-severe group in the univariable analysis. A prospective multicenter study of 2864 consecutive acute ischemic stroke patients in China reported that women aged >63 years were more likely to have intracranial artery stenosis than men [43]. This sex difference in the risk of intracranial artery stenosis is complex and not easily explained. However, elderly women have more vascular risk factors, such as DM, hypertension and dyslipidemia, than elderly men [43]. Additionally, elderly females are more likely to have hormone imbalance. Low sex hormone–binding globulin levels and high free androgen index are strongly associated with cardiovascular risk factors (DM, dyslipidemia and inflammation) in multiethnic premenopausal and perimenopausal women [44]. This could explain the sex difference for intracranial artery stenosis. However, in this study, multivariable logistic regression analysis adjusted for sex showed that SD, %CV, and MAGE were independent factors associated with severe ICA siphon stenosis, although there was a possibility that sex difference could have affected the tolerance of vessel structural change. 

Our findings support the idea that glucose fluctuation may help predict intracranial artery stenosis and accordingly direct preventive measures against ischemic stroke in T2DM patients. The rate of ischemic stroke episode was not significantly different between severe and non-severe ICA siphon stenosis. This may be due to the relatively small sample size of patients with severe stenosis or because patients with ≥80% carotid artery stenosis were excluded from the current study. However, identifying factors associated with severe intracranial artery stenosis is important because patients with severe intracranial stenosis have the highest rate of recurrent stroke [2,3]. Furthermore, glucose fluctuation is also associated with early neurological deterioration and poor functional outcome in patients with acute ischemic stroke [45]. Interventions for glucose fluctuation can prevent intracranial artery stenosis and ischemic stroke in T2DM patients.

This study has some limitations that need to be considered when interpreting the results. First, this was a cross-sectional study, and thus the causal association between glucose fluctuation and intracranial artery stenosis still needs to be clarified in studies with longer follow-up. Second, this study was conducted at a single center that was specialized for stroke and cardiovascular disease, and there was a relatively small number of patients with severe intracranial stenosis. Multicenter studies with a larger sample size are needed to further confirm the association between glucose fluctuation and ICA siphon stenosis. Third, patients with ≥80% carotid artery stenosis were excluded in this study because severe carotid artery stenosis is known to affect cognitive function [18]. This may lead to difficulty in interpretation of the results due to poor diabetes control of dementia patients. Glycemic variability is correlated with carotid IMT, which is an indicator of subclinical atherosclerosis [29]. Additionally, patients with intracranial artery stenosis tend to have carotid artery stenosis [46]. It is assumed that patients with ≥80% carotid artery stenosis have greater glycemic variability. It is therefore necessary to conduct future studies that include patients with carotid artery stenosis ≥80% along with a detailed neuropsychological assessment. Fourth, MRI may have lower accuracy than digital subtraction angiography (DSA) or computed tomography angiography (CTA) for evaluating stenosis. ICA siphon, where the stenosis was observed in this study, runs parallel to the axial images. Therefore, saturation of the blood signal may result in poor vessel delineation. However, some diabetes patients have chronic renal failure, which sometimes makes it difficult to perform DSA or CTA. Advances in vascular imaging technology are eagerly awaited.

## 5. Conclusions

Glucose fluctuation, as indicated by elevations in SD, %CV, and MAGE, is significantly associated with severe ICA siphon stenosis. Thus, glucose fluctuation can be a target of preventive therapies for intracranial artery stenosis and ischemic stroke.

## Figures and Tables

**Figure 1 nutrients-13-02379-f001:**
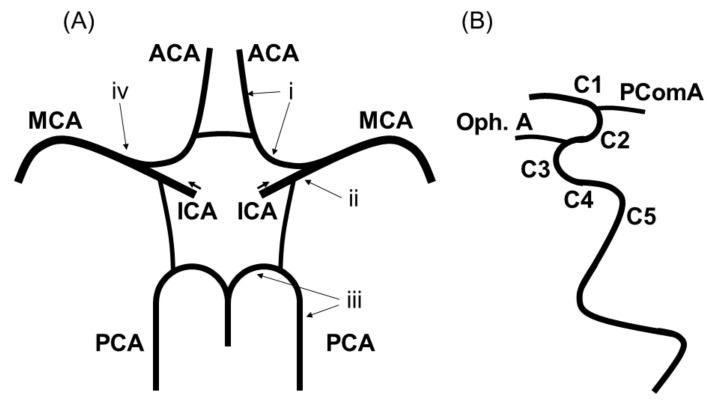
Evaluated vessels comprising the intracranial arteries. Schematic of the intracranial arteries evaluated in this study. (**A**) (i) The A1 or A2 segment of the ACA, (ii) the C1 to C5 segment of intracranial ICA, (iii) the P1 or P2 segment of the PCA, and (iv) the M1 or M2 segment of the MCA. (**B**) Classification of intracranial ICA according to Fischer’s classification: C1, from the ACA branch to the PComA branch; C2, from the proximal PComA branch to the ophthalmic artery branch; C3, from the ophthalmic artery branch to the genu of the internal carotid artery; C4, in the cavernous sinus; and C5, from the proximal cavernous sinus to the orifice of the carotid canal. Abbreviations: ACA, anterior cerebral artery; ICA, internal carotid artery; MCA, middle cerebral artery; Oph.A, ophthalmic artery; PCA, posterior cerebral artery; PComA, posterior communicating artery.

**Figure 2 nutrients-13-02379-f002:**
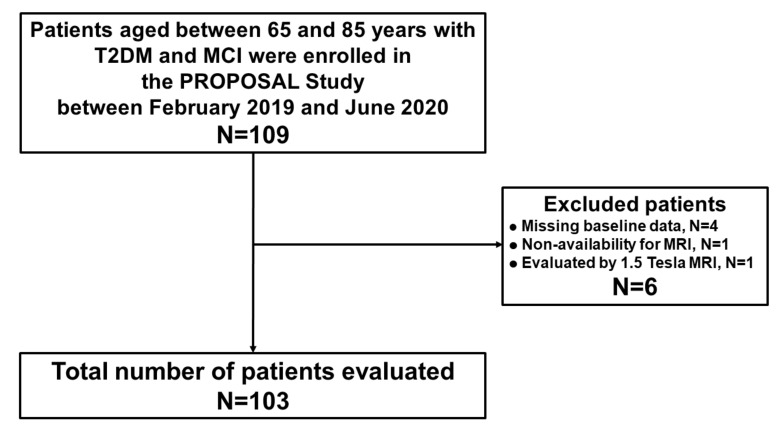
Patient inclusion flow chart. Blood glucose fluctuations were measured via CGM for at least 7 days in all patients. Abbreviations: CGM, continuous glucose monitoring; MCI, mild cognitive impairment; MRI, magnetic resonance imaging; T2DM, type 2 diabetes mellitus.

**Figure 3 nutrients-13-02379-f003:**
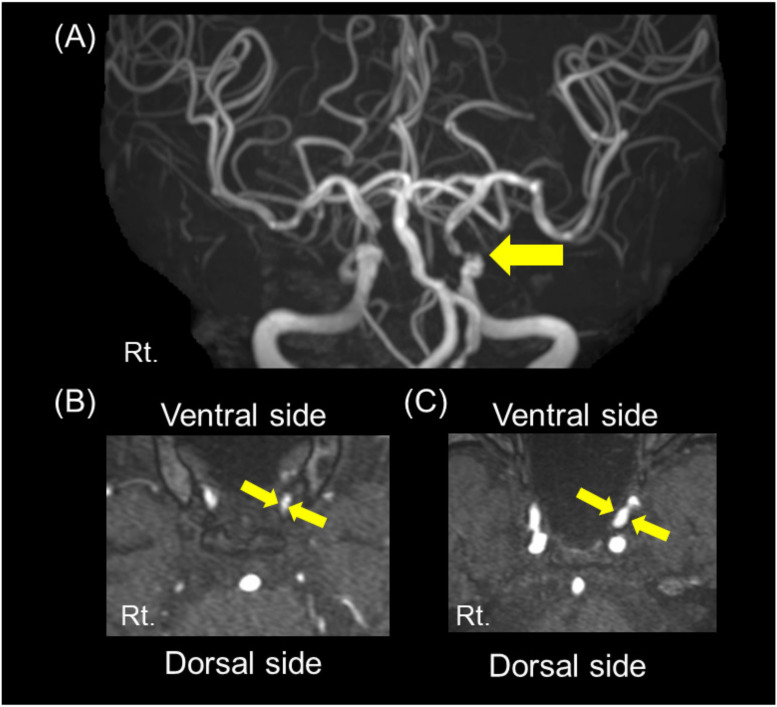
A representative case of severe internal carotid artery siphon stenosis. (**A**) Magnetic resonance angiography (MRA) showing the severe stenosis in the left internal carotid artery siphon (arrow). (**B**,**C**) MRA source images showing 74% stenosis of the internal carotid artery (ICA) siphon evaluated using the WASID method, with the narrowest portion (**B**, between arrows; 1.0 mm) of the siphon ICA and the widest portion (**C**, between arrows; 3.8 mm) of the petrous ICA.

**Figure 4 nutrients-13-02379-f004:**
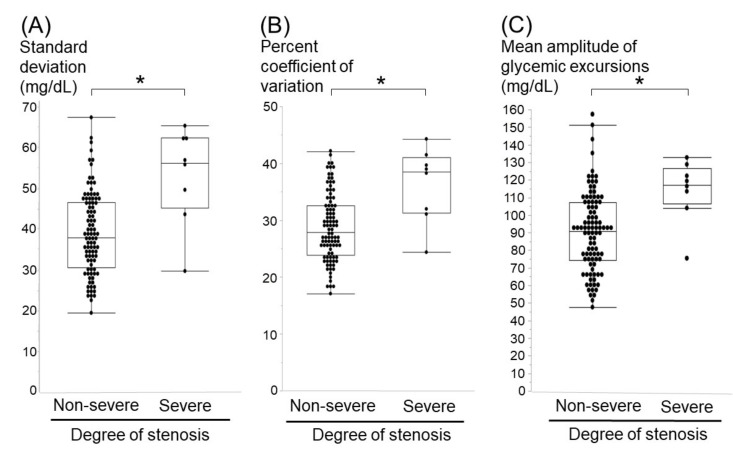
Scatter plots showing relationships between glucose fluctuation and internal carotid artery siphon stenosis. Glucose fluctuations assessed by standard deviation (SD) (**A**), coefficient of validation (%CV) (**B**), and mean amplitude of glycemic excursions (MAGE) (**C**) are significantly higher in the severe stenosis group than in the non-severe stenosis group (* *p <* 0.01).

**Table 1 nutrients-13-02379-t001:** Baseline patient characteristics by degree of internal carotid artery siphon stenosis.

	Intracranial Internal Carotid Artery Siphon Stenosis		
Variables	Severe (*n* = 8)	Non-Severe (*n* = 95)	*p* Value
Baseline demographics			
Female (%)	6 (75)	26 (27)	0.01
Age, years	76 ± 5	76 ± 5	0.63
Cerebrovascular risk factors			
Hypertension (%)	6 (75)	88 (93)	0.14
Dyslipidemia (%)	8 (100)	83 (87)	0.59
Current or past smoking (%)	3 (38)	54 (57)	0.46
Atrial fibrillation (%)	1 (13)	16 (17)	1.00
Medical history of percutaneous coronary intervention or coronary artery bypass grafting (%)	4 (50)	41 (44)	0.73
Ischemic stroke episode (%)	3 (38)	42 (45)	1.00
Diabetes mellitus			
Duration, years	26 ± 10	23 ± 11	0.32
HbA1c at registration, %	7.8 ± 0.9	7.5 ± 0.9	0.47
Blood glucose at registration, mg/dL	136 ± 42	150 ± 42	0.54
Blood glucose (CGM average), mg/dL	148 ± 25	136 ± 28	0.19
SD, mg/dL	53 ± 12	39 ± 10	<0.01
%CV	36 ± 7	29 ± 6	<0.01
MAGE, mg/dL	114 ± 18	90 ± 23	<0.01

Continuous variables are shown as the mean (± SD), while categorical variables are shown as frequencies and percentages. Abbreviations: HbA1c, hemoglobin A1c; CGM, continuous glucose monitoring; SD, standard deviation; %CV, coefficient of variation; MAGE, mean amplitude of glycemic excursions.

**Table 2 nutrients-13-02379-t002:** Multivariable analysis of influencing factors of stenosis adjusted for sex.

	SD (10 mg/dL)		%CV (/10)		MAGE (10 mg/dL)	
		*p* Value		*p* Value		*p* Value
Crude OR (95% CI)	3.60 (1.60–8.08)	<0.01	7.85 (1.90–32.5)	<0.01	1.56 (1.11–2.20)	0.01
Adjusted OR (95% CI)	3.00 (1.32–6.84)	<0.01	5.55 (1.23–25.2)	0.03	1.52 (1.06–2.19)	0.02

Abbreviations: SD, standard deviation; CV, coefficient of variation; MAGE, mean amplitude of glycemic excursions; OR, odds ratio; CI, confidence interval.

## Data Availability

The data presented in this study are available on request from the corresponding author.

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
