# Peer review of "Glucose Fluctuation and Severe Internal Carotid Artery Siphon Stenosis in Type 2 Diabetes Patients"

_nutrients, 2021, doi:10.3390/nu13072379_

Round 1

Reviewer 1 Report

In the present article authors have investigated the effect of glucose fluctuation on severe internal carotid artery stenosis. The present study provides novel data on essence of treating the glucose fluctuation efficiently and possibly preventing intracranial artery stenosis. Despite the low number of patients, and some limitations discussed in the last paragraph of discussion, present article is novel and well written.

Some minor comments:

Figure 2 would benefit from revision. At present form it is not so readable and informative.

A table presenting moderate <70% stenosis of ICA and also MCA stenosis (3 cases) could be valuable?

Reviewer 2 Report

The deteriorative role of glucose fluctuation in cardiovascular diseases is well recognized. Here, the authors aimed to analyze the correlation between glucose fluctuation and intracranial artery stenosis in type 2 diabetes patients with an aim in linking stroke. They concluded that glucose fluctuation is significantly associated with severe intracranial artery stenosis in T2DM patients. Thus, glucose fluctuation can be a target of preventive therapies for intracranial artery stenosis and ischemic stroke.

  1. This study is part of clinical study on the relationship between glucose fluctuation and cognitive function in T2DM. Patients with >80% carotid artery stenosis or other treatments were excluded by the original study. Can the exclusion criteria have impact on study conclusion?
  2. Among the 109 enrolled patients, only 8 patients are severe internal carotid stenosis and the remaining 95 are non-severe category. Sample size of eight is real small.
  3. Basically, the tolerance of vessel structural change may be of gender difference. Please have comment on it.
  4. According to statement, intracranial artery stenosis is of role in the incidence and severity of stroke. Although glucose fluctuation appears to have difference between severe and non-severe group, the ischemic episode is not different. Please have comment on it.

Round 2

Reviewer 2 Report

There is no additional comment.